# Medicine Shortages in Serbia: Pharmacists’ Standpoint and Potential Solutions for a Non-EU Country

**DOI:** 10.3390/pharmaceutics13040448

**Published:** 2021-03-26

**Authors:** Nataša Jovanović Lješković, Aleksandra Jovanović Galović, Svetlana Stojkov, Nikola Jojić, Slobodan Gigov

**Affiliations:** 1Faculty of Pharmacy, University of Business Academy, 21000 Novi Sad, Serbia; natasa.ljeskovic@faculty-pharmacy.com (N.J.L.); svetlana.stojkov@faculty-pharmacy.com (S.S.); nikola.jojic@faculty-pharmacy.com (N.J.); slobodan.gigov@faculty-pharmacy.com (S.G.); 2College of Vocational Studies for the Education of Preschool Teachers and Sport Trainers, 24000 Subotica, Serbia

**Keywords:** medicine shortages, non-EU country, Serbia, public pharmacies, hospital pharmacies

## Abstract

Medicine shortages in Serbia have evidently been present for several decades, but literature data are scarce. The aim of our study was to get an insight on the present situation in Serbia, review the EU actions when managing shortages, and discern a set of potential measures. A short survey was conducted among 500 pharmacists in public pharmacies, in 23 cities in Serbia. The survey questions addressed frequency of drug shortages, professional actions in the event of shortages, main consequences to patients and pharmacies, putative causes, and pivotal measures for the prevention/mitigation of drug shortages under current conditions. Moreover, a Panel of Experts was organized, whose suggestions and opinions were used to analyze the present situation and to form a set of potential solutions and effective measures to mitigate shortages of medicines. In-depth analysis of current Serbian legislation was conducted, with emphasis on specific steps to be made within the actual legal framework. Examples of good practice in the EU, applicable to a country such as Serbia, were examined. Our research showed that although Serbia is, in some aspects, behind EU countries regarding the approaches to overcome medicine shortages, progress can be made within short period of time, by specific well-targeted actions. Both patients and pharmacists would benefit from it.

## 1. Introduction

A drug shortage is complex, global phenomenon, occurring all over the World, recognized by the World Health Organization (WHO). It is a situation when the procurement of a particular medicine is not sufficient-temporarily or permanently—for a healthcare system’s needs. Shortages could be caused by problems in production, distribution, or by an increased demand. Problems in the production process, insufficient resources for drug procurement, or a sudden disease spreading are among the most frequent causes of medicine shortages [1,2].

Shortages of available drugs affect the healthcare system as a whole, including patients who may suffer the most serious consequences. Due to an insufficient supply or a complete lack of certain medications, patients do not receive appropriate healthcare, there is a higher possibility of medical errors, the recovery period is often prolonged, and healthcare expenses are higher [3]. Sudden and unexpected outbreaks of infectious diseases in certain world regions or globally can raise healthcare expenses enormously [4].

Although every drug could be affected by a shortage, the most often shortages occur for generic and less expensive drugs and parenteral forms of medicines. Some of these drugs are essential for healthcare system and patients, such as antibiotics, anticancer and analgesic drugs, vaccines, and especially orphan drugs [5,6,7,8]. Countries all over the world allocate high amounts of money in order to prevent and reduce drug shortages. On the other hand, healthcare professionals spend a significant amount of their working hours dealing with shortages.

Community and hospital pharmacists are on the front line of these problems. Community pharmacists have a significant role in taking care of a wide stratum of patients during the medicine shortage. They communicate with producers, distributors, and regulatory bodies in order to provide information about available drugs, and have consultations with medical doctors about alternative therapies and advise patients about it [9].

The aim of this study was to get an insight into the present situation in Serbia through an inquiry among professionals (pharmacists from public pharmacies and experts); relate the findings to current Serbian legislation, analyze examples of good practice in the EU; and discern a set of potential measures applicable to a non-EU country. Specific solutions were also discussed.

## 2. Materials and Methods

In order to achieve the aims of our study, we used a combination of qualitative and quantitative methods:a short survey was conducted among 500 pharmacists in public pharmacies in 23 cities in Serbia.a Panel of Experts was organized, whose suggestions and opinions were used to analyze the present situation and form a set of potential solutions and effective measures to mitigate shortages of medicines.in-depth analysis of current Serbian legislation was conducted, with emphasis on specific steps to be made within the actual legal framework.examples of good practice in the EU, applicable to a country such as Serbia, were examined.

### 2.1. Short Survey Conducted in Public Pharmacies

A survey among pharmacists employed in public pharmacies was conducted in 2019. A specially designed questionnaire consisting of 15 questions was developed in consultation with national and foreign experts, members of the European Medicines Shortages Research Network (COST CA15105). The questionnaire was distributed electronically (via Google Forms) to pharmacies statewide. Participation in this survey was voluntary and anonymous, and the Ethical Committee of Faculty of Pharmacy Novi Sad officially approved conduction of the study. Pharmacists permanently employed in public pharmacies in the territory of the Republic of Serbia were eligible to participate, without regard to gender, age, or years of professional experience. The survey was conducted in the territory of Serbia, in community pharmacies in Subotica, Sombor, Kikinda, Senta, Ada, Čoka, Kanjiža, Novi Kneževac, Bezdan, Sremska Mitrovica, Laćarak, Novi Bečej, Belgrade, Novi Sad, Vršac, Smederevo, Požarevac, Niš, Užice, Nova Varoš, Sjenica, Prijepolje, and Pančevo. The data collected from this survey were processed and presented graphically.

### 2.2. Panel of Experts

With the aim of investigating problems in the supply and distribution of medicines on the Serbian market from various standpoints, a shortened variant of the Delphi method-Panel of Experts-was used in the study [10,11]. Panel was organized with an emphasis on the exchange of opinions and experiences, but also as a ”scientific approach to consensus” [12]. Apart from the consensus on certain crucial points, one of the objectives was to disseminate relevant information and experience among the broader community of pharmacists and health professionals, and at the same time, to educate professionals on this important issue. To that end, the host of the Panel was the Chamber of Commerce and Industry of Serbia (CCIS), as an institution of substantial credibility and a long history (founded in 1857.). For the event titled “Health-risk assessment and a response to challenges of medicines shortages,” the invited speakers were hospital pharmacists [13], public pharmacists [14], and health professionals from wholesale companies and the pharmaceutical industry, while the moderators were experts in pharmaceutical legislation and from academia [13]. Apart from the invited panelists and moderators, more than 90 pharmacists from all over the country enrolled on the Skype for business platform.

Initially, the Panel of Experts was planned to be conducted as the final event of the shortened Delphi method in the form of a round table discussion at the Faculty of Pharmacy. However, the SARS-CoV-2 (Severe Acute Respiratory Syndrome Coronavirus 2) pandemic limited many activities in 2020. So, the planned event was organized, in collaboration with the CCIS, as an online expert discussion and an educational event for the pharmacists. The CCIS acknowledged the importance of this topic and the relevance of having the experts to discuss all important issues related to medicine shortages. Panelists were invited by email and informed about the panel topics.

## 3. Results and Discussion

### 3.1. Short Survey Conducted in Public Pharmacies

In order to gather relevant data directly from pharmacists on certain aspects of drug shortages in Serbia, a short 15-question survey was conducted in public pharmacies. The majority of the participants were women (470), belonging to three age groups (24–35; 36–45; >45), and had various levels of professional experience (0–10; 11–20; >20 years of experience), as depicted in Table 1.

The survey questions addressed the frequency of drug shortages, professional actions in the event of shortages, main consequences to patients and pharmacies, putative causes, and pivotal measures for the prevention/mitigation of drug shortages under current conditions. For the majority of questions (10 out of 15), there was one answer predominantly chosen by participants (≥50%), indicating the uniform attitude of pharmacists toward the posed questions (Figure 1, Figure 2, Figure 3 and Figure 4).

This study, conducted among pharmacists in Serbia, showed that medicine shortage is recognized as an important issue and that the attention it receives does not differ from the one in other countries of the region and broader. In the period (February–April 2019) when the survey was conducted, 98% of the participants were facing a shortage. Half of the pharmacists involved in the study stated that in the line of their work shortages arise once a week, and around a quarter (26%) once a month. These results are in accordance with the results from Germany [15], China [16], and the EAHP (European Association of Hospital Pharmacists) study conducted among hospital pharmacists in 2014, where 86% of the hospital pharmacists from 36 countries responded that they have a drug shortage problem in their own country weekly, sometimes daily [17].

The frequent steps undertaken by the pharmacists included transition to equally efficient therapy (68% of participants) and redirection of the patients to other health institution (18% of the participants), which is in line with the ASHP (American Society of Health-System Pharmacists) recommendations as part of the good communication and cooperation between health organizations [18].

More than half of the surveyed pharmacists (64%) thought that during a period of shortage, adequate functioning is possible although with difficulties, while 16% of the participants stated that functioning is impaired. Almost all of the pharmacists (94%) confirmed that in the event of a shortage, additional time is spent to gather information, identify an appropriate alternative, and communicate with other sectors in the supply chain. In a study from 2018 conducted among hospital pharmacists, it was found that the time required to deal with medicine shortages is five and a half working hours per week [19].

Only 8% of the pharmacists involved in our survey stated that the communication between interested parties is not adequate, which seemingly implies a certain advantage in comparison to a study conducted by the U.S. Institute for Safe Medication Practices (ISMP) from 2010 to 2017. According to these investigations, 84% of the respondents during 2010 stated that they had never or rarely received information in advance about shortages, while this percentage was lowered to 38% in the year 2017 [20].

The surveyed pharmacists in our investigation declared that shortages on a national level may be connected with the inability of registering medicines within the period of three months (42%) or six months (36%). It is worth mentioning that according to the current legislation in Serbia, the Medicines and Medical Devices Agency of Serbia (ALIMS) has a period of 60 days to determine the validity of the request for registration, and an additional 210 days for the decision about a registration license, which equals a period of approximately nine months in total.

According to the opinion of most respondents (62%), drug shortages have become an issue in the last decade or more, which confirms the results of numerous studies in other countries: The situation in the USA became critical in 2011, when the number of reported shortages reached 267 [21]. A rise in the number of shortage events in the second decade of the 21st century has been recorded in the data issued by the EMA (European Medicines Agency), which reported 438 shortages during 2014 vs. 44 during the year 2008 [22]. In the three months during our short survey (February–April 2019), among the pharmacists from public pharmacies, shortages of 36 medicines were detected. Half of the respondents had the perception that the problem has aggravated in the last several years, 40% did not perceive any significant change, while 10% of the pharmacists thought that improvements have been made.

As the principal cause for shortages in Serbia, participants identified the duration of the preregistration process (44%) and the registration of medicines (24%), as well as the procurement of raw materials (24%). In studies conducted in various European countries (Italy, France, Germany, Spain, United Kingdom, Netherlands, and Belgium), administrative procedures have rarely been recognized as a cause for shortages, with the exception of Italy. Shortages of a raw materials/manufacturing problems have been recorded in the majority of the countries involved in this study, while as economic factors, most frequently listed were commercial and financial issues (Italy, Netherlands, and Belgium) [1]. As per economic factors, 40.4% of the participants in our investigation stated cost-effectiveness on the market. In addition, stringent quality control and GMP (Good Manufacturing Practices) standards were reported (30.3%). Regarding the factors that limit possibility of improvement in drug supply, half of the participants stated the duration of registration procedures, while around a quarter (26%) thought that the legislative framework should be changed.

### 3.2. Panel of Experts

#### 3.2.1. Management of Medicine Shortages in the EU and Examples of Good Practice

Medicine shortages are recognized as an increasingly frequent problem in the EU [23]. Directive 2001/83/EC, Article 23a, clearly states a requirement for all stakeholders to notify national regulatory bodies two months in advance about anticipated temporary or permanent halts in the supply [24]. However, according to the European Medicines Agency (EMA) research in 2015, this regulation is observed in 21 out of 28 member countries. Of note is the fact that 50% of all shortages involve oncological, antimicrobial, and drugs for CNS (Central Nervous System) diseases. EU drug production is highly dependent on foreign—mainly Asian—industry, since 80% of active principles (APIs) and 40% of all drugs are produced in India and China. Penicillin EU requirements almost entirely (90%) depend on Asian production. Indeed, there are efforts to translocate API production from Asia to Europe, expressed also through the EU Parliament recommendations. The EU strategy for Pharmacy (adopted in December 2020) also emphasizes consolidation of production. In the same document, accessibility of innovative therapeutic options and stringent control in the price augmentation of medicines are listed as priorities. Where medicine shortages are concerned, coordinated actions are advocated for in many EU documents (EU Parliament recommendations, EMA documents, etc.). In a report of the European Parliament’s Committee on Environment, Public Health and Food Safety (approved by the EU Parliament in September 2020), a set of measures to be taken in response to the pandemic was listed. Emphasis was placed on the establishment of a coordinated plan of action for the prevention and management of drug shortages. To that end, a guide for the implementation of MEAT (monitoring, evaluation, assessment, and treatment) has been published. Among the measures there is the establishment of a digital platform for real-time notification on shortages, composition of a functional catalog of drugs in shortages (managed by the EMA), new strategies for stock management, and procurement from more than one supplier.

The EMA/HMA (Heads of Medicines Agencies) Action Group monitors drug accessibility and emphasizes the importance of transparent information exchange, and shortages have to be communicated among stakeholders in the right way. EMA—SPOC (Single Point of Contact) is an innovative data exchange system among EU regulatory bodies. A variant of this system is iSPOC (Industry Single Point of Contact), by which pharmaceutical industries report problems in drugs availability directly to the EMA and national regulatory bodies.

The EMA Risk Assessment Indicators for drug shortages encompass 11 indicators, among which are drug characteristics that could prevent or influence therapeutic substitution, patient health risks in the onset of a drug shortage, and preparation time for patients. A prospective risk assessment is a valuable approach [25,26,27] and is essential for planning.

Nevertheless, current EU catalogs of shortages rarely offer available alternative therapies, which is a weakness that should be corrected.

As an example of good practice, it is worth mentioning the Netherlands, which established a centralized database for medicines in shortage. This multi-stakeholder approach was launched in 2004. and in 2018. the official web site had one million visits. The main reason for its popularity and influence lies in the fact that it offers verified information and is completely solution-oriented (contains a list of alternative drugs; possibility for magistral medicines and expected duration of shortage; possibility for import; causes of shortage, etc.) [28].

Innovative approaches also include software design and digital tools that can predict the onset of a drug shortage. These solutions go a step forward, enabling timely and coordinated mitigation measures conducted by all involved subjects. Horizon scanning to identify “programmed obsolescence” and budgeting is an example of an approach already implemented in Sweden [29].

The situation in the EU countries inevitably has significant impact on the Serbian pharmaceutical market. There is no ideal, single approach or model that could lead to a solution, and models developed in the EU cannot simply be copied and applied to Serbia. Rather, bits and pieces of novel approaches may be useful if carefully selected and adjusted to the circumstances in a specific country outside of the EU. Nevertheless, a comprehensive overview of all medicine shortages together with a list of possible solutions, like the one provided by The Royal Dutch Pharmacists Association (https://farmanco.knmp.nl/ accessed on 15 January 2021), may be a useful tool anywhere, including Serbia, enabling pharmacists, medical doctors, and patients to timely and effectively react to medicine shortages. To a non-EU country important take-home messages from all EU documents are: networking and transparency, real-time flow of information, action plan at onset of shortage, and coordinated actions of all stakeholders.

#### 3.2.2. Regulatory Framework in Serbia—A Short Overview

According to the Law on Health Protection in the Republic of Serbia [30], pharmacies have an obligation to supply medicines to the general public, health institutions, and other legal entities. Procurement of industrially produced drugs depends also on other stakeholders, among which of major importance are the Ministry of Health, the Medicines and Medical Devices Agency of Serbia, and the National Health Insurance Fund. Our analysis of public and hospital pharmacy practice in the case of drug shortages showed that there is an obligation to report shortages on a weekly basis to the Ministry of Health and the National Health Insurance Fund. The current legal framework gives pharmacists a possibility to substitute prescribed therapy in the case of a shortage, with an adequate generic drug (same composition, formulation, dosage, packaging, and price) [31,32], while alternative therapy is not legally defined. Furthermore, in the event of shortage, pharmaceutical and health institutions have a legal right to suggest the import of a certain drug to the relevant company (importer), which has to be subsequently forwarded to the Medicines and Medical Devices Agency of Serbia [33]. It is important to note that this right is not available to independent pharmacies, who could provide an important contribution in drug shortage detection and management. Moreover, experience has shown that there is a need to simplify and shorten this procedure, as well as to reduce the expenses to the legal entity that suggested the import of a drug [34]. Since Serbia is in the process of harmonization of legal acts with the European Union (EU), it would be reasonable to recognize EU certificates for the release of a drug on the Serbian market in case of an import from an EU country, which is not the case at present.

Analyzing the legislative framework that deals with the supply of the pharmaceutical market, primarily the Law on Health Care and the Law on Medicines and Medical Devices, we conclude that this area is well regulated and generally harmonized with laws of neighboring countries, still legal solutions are not always and completely reflected in practice.

The principles of comprehensiveness and continuity of healthcare are defined by the Law on Health Care (Articles 22 and 24) and are the responsibility of all relevant institutions, organizations, and individuals. The Law on Medicinal Products and Medical Devices defines the specific obligations of the holder of a medicinal product license to continuously supply the market (Article 132) and including the one to inform the competent ministry without delay about any problem of continuous supply (Article 139). Violations of the above provisions are among the economic offenses for which fines are determined, yet the authors are not aware that any such sanctions have been implemented in the previous period.

#### 3.2.3. Medicine Shortages from the Perspective of Public Pharmacies in Serbia

The apothecary sector at the primary level in Serbia is in the process of transition from state-owned toward privately owned business. Changes, brought by the reform of health services, have struck pharmacies in an unprepared state. State pharmacies—monopoly owners on prescription drugs for many decades—were not prepared for the competition on the free market, while privately owned pharmacies were not ready for complex tasks that they were expected to fulfill as public health stakeholders. Practice has shown that even the legal framework, which was supposed to give legal support for the transition of the whole network of health institutions in Serbia [35], was not sufficiently adjusted to the ongoing changes, which additionally slowed down the development of the apothecary sector and did not contribute to solving numerous problems.

Even if the transition process had occurred in a systematic and gradual way, which was not the case, this change would not have been an easy process. In reality, public pharmacies have experienced a “tectonic“ change, with many of them being sold to private pharmacy chains, which was reflected—among other issues—on medicine shortages.

Channels through which drugs are brought onto the market in Serbia encompass various stakeholders—from producers, license bearers, and wholesalers to pharmacies. A large set of legal documents regulate this area. The number of wholesale companies in recent years has significantly changed in relation to the type of ownership (since all companies with state ownership ceased to exist) and in relation to the type of drug distribution model (wholesalers vs. direct sellers). The current situation, with only a few wholesalers doing business on the market, has led to a large increase in profit margins caused by close-to-monopoly conditions. Thus, it should not be surprising that all the links in the chain of drug supply are not always working in harmonious way. Pharmacies are positioned at the end, thus bearing most of the burden (apart from the patients) imposed by medicine shortages [36,37,38].

Pharmacies also have a special role in the supply of drugs, which they perform through magistral and galenical production. As health professionals competent in drug manufacturing, pharmacists may significantly contribute to the resolution and management of shortages by producing certain drugs in laboratories at pharmacies. This practice is present in Serbia and is regulated by the Law on Medicines [39], and by numerous legal acts dealing with the galenical production of medicines [40,41]. According to the Law on Medicines, a magistral drug is a drug produced in a pharmacy, by the formulation specified for a single patient (or client), while a galenical drug is a drug produced on the basis of current pharmacopoeia or magistral formulations in a galenical laboratory and is dedicated to patients in series of 300 single packages. The development of a galenical product is somewhat difficult due to the fact that galenical laboratories, located in pharmacies, cannot produce parenteral preparations and cannot supply secondary-level health institutions (e.g., hospitals), in spite of the fact that no such institution in Serbia has a registered galenical laboratory.

Even though there are obvious difficulties, there are examples where the entrepreneurial spirit of pharmacists has overcome some of the shortages of medicines on the market. This is evident especially in specific market niches, such as dermatological and ophthalmological products, specific formulations (vaginal tablets, suppositories, enemas, etc.), as well as personalized dosages and forms of drugs. However, there is a problem in the reimbursement from the Republic Fund for Health Insurance when magistral drugs are concerned. The reimbursement is of special importance for socially vulnerable groups, for which such drugs are frequently indicated (e.g., emulsion of benzyl-benzoate in the treatment of scabies).

#### 3.2.4. Medicine Shortages from the Perspective of Hospital Pharmacies

In 2018 and 2019, the European Association of Hospital Pharmacists (EAHP) published the results of a survey on drug shortages in the hospital sector. A survey indicated that, despite efforts to counteract shortages, almost 92% and 95% of hospital pharmacists, respectively, reported experiencing shortages. The types of medicines most frequently in shortage were antimicrobial agents (63%), oncology medicines (47%), and anesthetic agents (38%) [42].

In recent years, difficulties with drug shortages in hospital pharmacies are being resolved by using alternative ways of treatment administration, bearing in mind patient benefits. According to the information provided at the Panel of Experts, in 2020, a shortage of anticancer biologics was detected on several occasions, and with the application of alternative pharmaceutical formulation of the same drug, successfully managed. For instance, periodic reports of critical drug shortages in Serbia confirmed a shortage of the intravenous form of trastuzumab, the first targeted therapy for human epidermal growth factor receptor 2 (HER2)-positive breast cancer. To insure that patient care is not interrupted, a switch to other pharmaceutical forms, in this case from intravenous to subcutaneous dosing, offered shortage resolution. Another example of the same approach was documented during our survey. In addition to trastuzumab, another anticancer biologic drug was deficient in 2020. The intravenous form of rituximab, a key component of most therapeutic regimens used for chronic lymphocytic leukemia and non-Hodgkin’s lymphoma therapy, due to shortage, was replaced by the subcutaneous form of the same drug, enabling rituximab efficacy to be maintained. However, on several occasions, even this transition from one pharmaceutical form to another was not possible due to legislative restrictions. For instance, in 2020, a shortage of the intravenous form of 5-fluorouracil was reported. Applying the above-mentioned approach, the intravenous (IV) form of fluorouracil could be managed with oral capecitabine, a prodrug for 5-fluorouracil. Eventually, switching between drugs could resolve the issue, but only in the case in which both drugs have the same indications approved by the National Health Insurance Fund. Unfortunately, this is not always the case. According to indications approved by the National Health Insurance Fund, 5-fluorouracil is indicated in the treatment of breast cancer, gastric cancer, and colorectal cancer. On the contrary, capecitabine is indicated only for breast and colorectal cancer, not being indicated for the treatment of gastric cancer. Thus, in a scenario where a patient with gastric cancer is facing a 5-fluorouracil shortage, no oral capecitabine could be introduced. In other words, the indications approved by the National Health Insurance Fund are not the same as the indications found in the summary of product characteristics, making the alternative pharmaceutical formulation strategy unavailable [43,44].

Although some drug shortages can be eased by changing pharmaceutical formulation, regimens that use paclitaxel often lack equally effective substitutes, adversely affecting the prognosis of many patients [45]. In the Panel of Experts, the participants from hospital pharmacies reported paclitaxel shortage, but an alternative administration route strategy could not be employed. Paclitaxel shortage was reported two times during 2020, in March and September. In these cases, a lack of paclitaxel impacted patient care in several ways. Having in mind that paclitaxel is often part of a multidrug combination, scarcity of paclitaxel resulted in the delay in care, leading to longer waiting periods for some patients, although other drugs used in combination with paclitaxel were available. Moreover, chemotherapy is often used in conjunction with surgery and radiotherapy, both being delayed due to the unavailability of chemotherapy.

In line with this, another barrier in successful navigation of a drug shortage is the lack of an integrated IT system in Serbia, supplied with information on ongoing and future shortages [46]. A combination of the lack of information and the re-occurring shortages resulted in the situation where both patients and healthcare professionals were faced with difficult decisions-either treatment interruption or a transition to less-effective/more expensive treatments. Of note is the fact that hospital pharmacists are obliged to daily report on ongoing drug shortages both to the National Health Insurance Fund and the Ministry of Health.

Another aspect with adverse impact on patient health is the fact that even the large healthcare institutions in Serbia do not have multiple suppliers for each drug, making the hospital pharmacy market even more vulnerable to drug shortages. Although hospital pharmacists have proposed changes in legislation, suggesting supply from various sources and multiple distributors to maintain a continual availability of drugs, the legislation changes take a long time. Moreover, there is no regulation that would force distributors to inform other parties in advance about potential shortages, although truth be told, in some instances even distributors do not have accurate information about the causes and duration of a drug shortage.

#### 3.2.5. Manufacturing Issues

In terms of manufacturing issues, the reasons behind medicine shortages can be divided into two main categories: The shortage of direct materials and the shortage of indirect materials. A shortage of direct materials compromising the availability of medicines is the inability of manufacturers to obtain raw active pharmaceutical ingredients, other ingredients such as drug excipients, or drug packaging/pharmaceutical packaging material. Having in mind that raw material shortages are frequently considered a global issue, manufacturers supplying the Serbian market are no exception [47]. Indirect materials being the reason behind medicine shortages accounts for the scarcity of personal protective equipment for the manufacturing sector or chemicals used for maintaining the cleanliness of the production lines. Another highly important indirect factor is the shortage of machinery maintenance workers based in Serbia; thus, in case of industrial machinery malfunction, crucial time is lost waiting for maintenance workers to come to Serbia. In the case of the SARS-CoV-2 pandemic, a deficit in the qualified workforce and border crossings have become critical points for adequate drug supply. According to Serbian legislation, all manufacturers supplying the local market are obligated to notify in advance the national authorities, for instance, the Ministry of Health, in case of potential or forthcoming drug shortages.

#### 3.2.6. Wholesaler Issues

Wholesalers as the vital bridge between pharmaceutical manufacturers and healthcare providers have a key role in preventing and managing drug shortages. The main efforts that wholesalers in Serbia make in the prevention of supply disruptions are improvements in communication, monitoring, and planning processes. In terms of communication, both internal communication between wholesales departments and external communication with regulatory bodies, customs, and end-users that supply medicines to the public have a crucial effect on mitigating the impact of drug shortages on providers and patients. One example presented by the wholesaler representatives on the Panel of Experts showed the impact of information flow. After being notified by the manufacturers in Italy that due to SARS-CoV-2 pandemic, a potential supply disruption could occur, wholesalers immediately notified the Ministry of Health and offered an alternative solution. After notifying the regulatory bodies, reliable information on the behalf of wholesalers was sent to the healthcare providers about the alternative pharmaceutical form for the same INN (International Nonproprietary Name), which can mitigate the impact that product shortages could have on patient care. Another strategy employed by wholesalers is weekly or monthly pharmaceutical market monitoring. For instance, wholesalers in Serbia monitor the fulfillment of requests from healthcare providers. This includes both healthcare requests regarding drug assortment and the number of drugs requested. This strategy allows pharmaceutical wholesalers to predict sales well in advance and to respond adequately in the case of drug shortages. In addition, approach wholesalers in Serbia readily employ is obtaining a buffer storage. For instance, in cases when a drug sales license is about to expire, along with the application procedure for obtaining a renewed sales license, a buffer storage is planned for a period of at least three to six months.

#### 3.2.7. Small Steps in Advance—Potential Solutions for a Non-EU Country

For the Non-EU Countries, the transformation journey aimed at tackling the problem of drug shortages follows a set of small steps, the first one probably being the harmonization of its legislation with the EU. Drugs registered in the European Union (EU) for human use are not automatically registered in the Republic of Serbia. Although the Republic of Serbia is not a member of the EU, there are examples when countries who were candidates for membership of the EU accepted registration of drugs in the EU and such drugs were automatically registered in candidate countries, which significantly improved the supply of the drug market, reducing the number and duration of drug shortages [48,49].

This practice could be adopted by the Republic of Serbia, which would help the healthcare system to obtain certain essential medications for patients when these drugs are in shortage in Serbia for a long period of time [50]. Furthermore, mutual recognition and registration of drugs that are already registered in the EU would have significant impact on the availability of orphan drugs [51].

When a drug shortage occurs in the Republic of Serbia, at present, there are no possibilities for the import of this drug from the EU; instead, they could be registered on the so called “D list”, with special prescribing and dispensing requirements. This special prescribing would be possible only for the medications that are essential for a certain patient or a group of patients and in cases when such drugs do not have a therapeutic parallel on the Serbian drug market. In these cases, a drug that is not registered in Serbia could be imported and registered and dispensed on D list through a shortened procedure. It is important to note that some of these drugs are used for the treatment of serious conditions (e.g., tuberculosis, AIDS, and epilepsy), and in some instances, they are on D list for years. It remains unclear why these drugs are not registered in Serbia by a regular procedure.

In the Serbian legislation, there are two forms of community pharmacies (healthcare facilities and private practices), but only healthcare facilities are allowed to propose the urgent import and registration of drugs on D list [52].

Since there are no majority state-owned pharmaceutical companies (producers) anymore in the Republic of Serbia, health authorities lack this important mechanism to directly stimulate the production of certain drugs that are in shortage. In this context, magistral and galenical manufacturing is of special importance. However, this approach raises other issues in terms of quality assurance and safety standards, which would need to be harmonized with EU regulations in the future [53].

So, in our view on the level of public pharmacies, there are two main approaches to solve the issues related to shortage of medicines:to enable interventional import of non-registered medicines;to prepare medicines in pharmacy laboratories (magistral and galenical manufacturing) and, thereafter, to dispense them to patients with a prescription from a medical doctor.

In addition, it would be essential to reinforce communication with medical doctors in order to implement proactive measures for overcoming medicine shortages at the primary care level. These may include prescription of: galenical/magistral manufacturing of medicines, another dosage form, another medicine from the same pharmacotherapy group, or alternative therapies. Furthermore, it would be important to have available a regularly-updated database containing information on medicine shortages (there is a reporting system of the Ministry of Health, but so far, it is still a one-way communication system, i.e., there is a need to develop a system of feedback on sent reports or an actual and transparent database that summarizes information on medicine shortages, causes, expected dates, alternative dosage forms and therapies).

## 4. Conclusions

In order to find solutions for medicine shortages, it is necessary to gain clarity of the root causes of the problem, which is not an easy task, given the fact that medicine shortages lie on the crossroads of at least five separate human disciplines—science, medicine, economy, politics, and law—each of which has its own specific rules and driving forces. Although widespread, drug shortages in various countries have their own specificities. If we strive to achieve a global solution, it should encompass all specificities and be flexible enough to provide aid for everyone on the planet.

Our study has given indications that, for Serbia, the main amendments should be made in the area of shorter, more effective procedures in the registration of imported drugs, better communication of all parties involved, and a greater degree of transparency of the procurement system. Backup manufacturing on a small scale (magistral and galenical) could be a good way to overcome some shortages. Furthermore, a long-term strategy could include support for the local production of generic drugs—in the form of economic and legislative incentives—which may be part of the solution, since many generic drugs do not have an adequate parallel, thus forcing patients to use alternative drugs, sometimes with lower efficacy or new side effects. The issue of medicine shortages requires a well-structured approach agreed by all stakeholders—pharmacists, medical doctors, wholesale companies, pharmaceutical companies, patient associations, insurance funds, medicine agencies, ministries, and other government institutions—in order to establish an efficient, effective, and sustainable solution to prevent and, when necessary, overcome shortages of medicines.

## Figures and Tables

**Figure 1 pharmaceutics-13-00448-f001:**
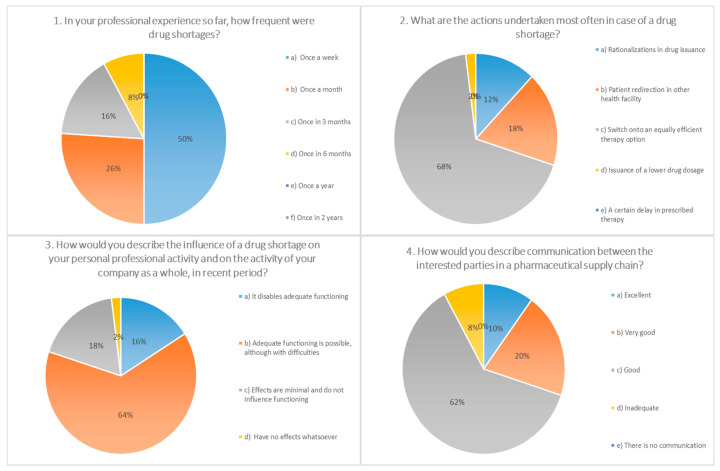
Questions 1 to 4 from a 15-question survey, distributed electronically to community pharmacies in 23 cities in Serbia on behalf of the Faculty of Pharmacy, Novi Sad. The survey was available for completion between February and April, 2019. The survey was anonymous, as contact information was not requested. Data collected from the survey included demographic information. The majority of the questions were multiple choice, allowing respondents to select a pre-specified answer. A total of 500 surveys were administered to community pharmacies.

**Figure 2 pharmaceutics-13-00448-f002:**
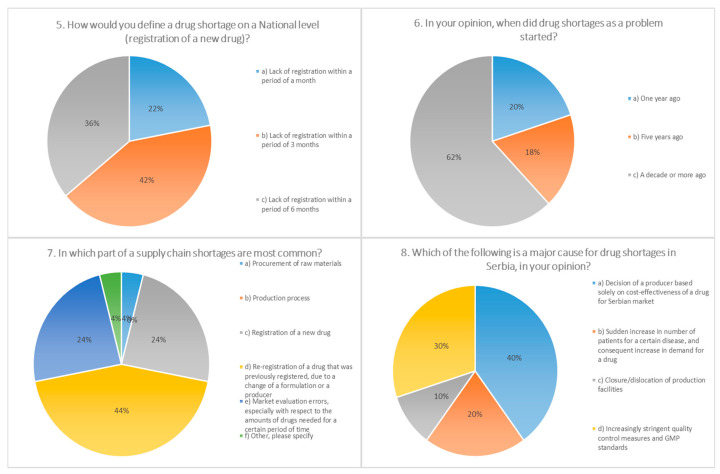
Questions 5 to 8 from a 15-question survey, distributed electronically to community pharmacies in 23 cities in Serbia on behalf of the Faculty of Pharmacy, Novi Sad. The survey was available for completion between February and April, 2019. The survey was anonymous, as contact information was not requested. Data collected from the survey included demographic information. The majority of the questions were multiple choice, allowing respondents to select a pre-specified answer. A total of 500 surveys were administered to community pharmacies.

**Figure 3 pharmaceutics-13-00448-f003:**
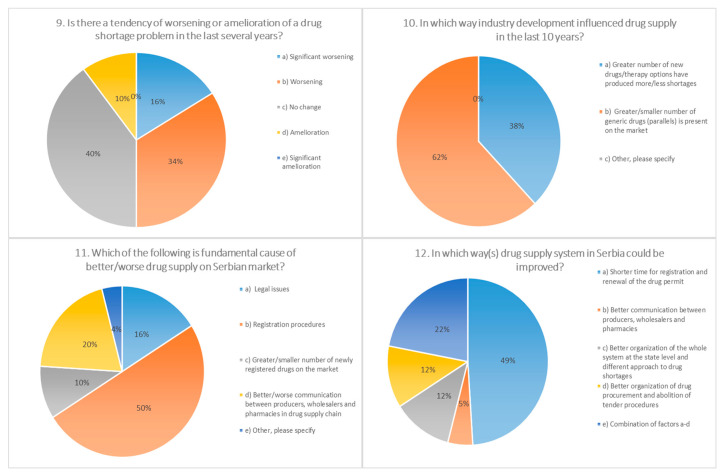
Questions 9 to 12 from a 15-question survey, distributed electronically to community pharmacies in 23 cities in Serbia on behalf of the Faculty of Pharmacy, Novi Sad. The survey was available for completion between February and April, 2019. The survey was anonymous, as contact information was not requested. Data collected from the survey included demographic information. The majority of the questions were multiple choice, allowing respondents to select a pre-specified answer. A total of 500 surveys were administered to community pharmacies.

**Figure 4 pharmaceutics-13-00448-f004:**
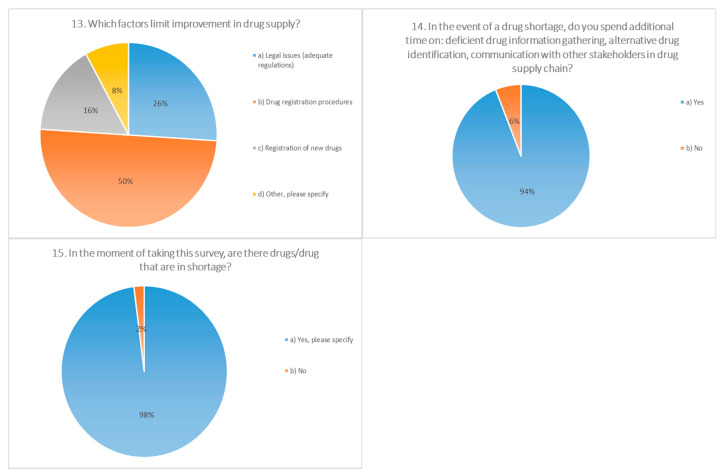
Questions 13 to 15 from a 15-question survey, distributed electronically to community pharmacies in 23 cities in Serbia on behalf of the Faculty of Pharmacy, Novi Sad. The survey was available for completion between February and April, 2019. The survey was anonymous, as contact information was not requested. Data collected from the survey included demographic information. The majority of the questions were multiple choice, allowing respondents to select a pre-specified answer. A total of 500 surveys were administered to community pharmacies.

**Table 1 pharmaceutics-13-00448-t001:** Gender, age group distribution, and years of professional experience of the participants in the short survey conducted in public pharmacies in 23 cities in Serbia, from February to April of 2019.

**Gender**	Women	Men		Total
	470	30		500
**Age group**	24–35	36–45	>45	Total
	180	120	200	500
**Experience-years**	0–10	11–20	>20	Total
	190	130	180	500

## Data Availability

Not applicable.

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
