# Peer review of "Medicine Shortages in Serbia: Pharmacists’ Standpoint and Potential Solutions for a Non-EU Country"

_pharmaceutics, 2021, doi:10.3390/pharmaceutics13040448_

Round 1

Reviewer 1 Report

This manuscript describes a particular situation in Serbia. The work presents a overview the problem in the shortage of medicines, and covers the main circumstances that cause the shortage.

However, the objective of the work is not clear, and the regularatory and legal aspects and the implication of the pharmaceutical industry should be more widely discussed.

The authors should emphasize that their work contributes.

Author Response

Dear Reviewer,

thank you for your comments and suggestions. Please find attached a file containing all our answers.

Yours Sincerely,

Aleksandra Jovanovic Galovic, PhD

Reviewer 2 Report

Pls see the attached comments 

Author Response

(The authors gave the same response as above.)

Reviewer 3 Report

The manuscript entitled Medicine Shortages in Serbia aims to provide to the readers the present situation in Serbia and to review the EU actions to manage shortages and present a set of potential measures.

The article is interesting and I believe it could be attractive for the readers of Pharmaceutics. 

Before to be considered for publication I have just a few comments:

1) The introduction part is interesting and well organized but I suggest to focus more on the situation in Serbia to be consistent with the title.

2) section 3.1 from line 165 to 169. I suggest moving this part to the methodology section

3) The Figures quality has to be improved, is very difficult to read.

4)section 3.1 : the results should be better discussed and supported by other studies related to the situation in Serbia, even performed time ago

5) line 368-369 : the authors clais that producing certain drugs in loco, in the laboratory located inside a pharmacy, could significantly contribute in resolving the management of shortages. My question is : considering the dimension and the equipment suitable for a standard laboratory located in an urban pharmacy, how it would impact on the final cost of the medicine? could be suitable for which categories of drugs? with the term producing a certain drug, it refers to the production of the final formulation? because in this case is only a question of assembly.

Author Response

(The authors gave the same response as above.)

Round 2

Reviewer 1 Report

The manuscript has been improved, its publication is suitable